# Impact of Functional Polymorphisms on Drug Survival of Biological Therapies in Patients with Moderate-to-Severe Psoriasis

**DOI:** 10.3390/ijms24108703

**Published:** 2023-05-12

**Authors:** Cristina Membrive-Jiménez, Cristina Pérez-Ramírez, Salvador Arias-Santiago, Antonio Giovanni Richetta, Laura Ottini, Laura Elena Pineda-Lancheros, Maria del Carmen Ramírez-Tortosa, Alberto Jiménez-Morales

**Affiliations:** 1Pharmacogenetics Unit, Pharmacy Service, University Hospital Virgen de las Nieves, Avenida. de las Fuerzas Armadas 2, 18004 Granada, Spain; cristinamembrive@correo.ugr.es (C.M.-J.);; 2Department of Biochemistry and Molecular Biology II, Faculty of Pharmacy, Campus Universitario de Cartuja, University of Granada, 18011 Granada, Spain; 3Dermatology Service, University Hospital Virgen de las Nieves, 18014 Granada, Spain; 4Unit of Dermatology, Department of Internal Medicine and Medical Specialties Sapienza, University of Rome, 00161 Rome, Italy; 5Department of Molecular Medicine, Sapienza University of Rome, 00161 Rome, Italy

**Keywords:** adalimumab, etanercept, infliximab, ustekinumab, anti-TNF, TLR5, HLA-C, TNF-1031, PDE3A, CD84

## Abstract

Biological therapies (BTs) indicated for psoriasis are highly effective; however, not all patients obtain good results, and loss of effectiveness is the main reason for switching. Genetic factors may be involved. The objective of this study was to evaluate the influence of single-nucleotide polymorphisms (SNPs) on the drug survival of tumor necrosis factor inhibitors (anti-TNF) medications and ustekinumab (UTK) in patients diagnosed with moderate-to-severe psoriasis. We conducted an ambispective observational cohort study that included 379 lines of treatment with anti-TNF (*n* = 247) and UTK (132) in 206 white patients from southern Spain and Italy. The genotyping of the 29 functional SNPs was carried out using real-time polymerase chain reaction (PCR) with TaqMan probes. Drug survival was evaluated with Cox regression and Kaplan–Meier curves. The multivariate analysis showed that the *HLA*-C rs12191877-T (hazard ratio [HR] = 0.560; 95% CI = 0.40–0.78; *p* = 0.0006) and *TNF*-1031 (rs1799964-C) (HR = 0.707; 95% CI = 0.50–0.99; *p* = 0.048) polymorphisms are associated with anti-TNF drug survival, while *TLR5* rs5744174-G (HR = 0.589; 95% CI = 0.37–0.92; *p* = 0.02), *CD84* rs6427528-GG (HR = 0.557; 95% CI = 0.35–0.88; *p* = 0.013) and *PDE3A* rs11045392-T together with *SLCO1C1* rs3794271-T (HR = 0.508; 95% CI = 0.32–0.79; *p* = 0.002) are related to UTK survival. The limitations are the sample size and the clustering of anti-TNF drugs; we used a homogeneous cohort of patients from 2 hospitals only. In conclusion, SNPs in the *HLA-C*, *TNF*, *TLR5*, *CD84*, *PDE3A*, and *SLCO1C1* genes may be useful as biomarkers of drug survival of BTs indicated for psoriasis, making it possible to implement personalized medicine that will reduce financial healthcare costs, facilitate medical decision-making and improve patient quality of life. However, further pharmacogenetic studies need to be conducted to confirm these associations.

## 1. Introduction

Psoriasis is a chronic autoimmune skin disease with a prevalence of 2–4% in Europe and the United States [1]. It is characterized by the development of lesions in the form of erythematous plaques with silvery-white scales, mainly on the scalp, elbows, knees and back [2,3]. Moreover, it is associated with other potentially disabling diseases [4,5]. Consequently, psoriasis is considered a systemic disease that has a major impact on patients’ quality of life and generates high healthcare costs [6].

Genetic, immunological and environmental factors can trigger psoriatic lesions, which are maintained by abnormalities in cutaneous immune responses. Activation of plasmacytoid dendritic cells, keratinocytes, natural killer cells and macrophages that secrete cytokines such as interferons beta and gamma IFN-β and IFN-γ), interleukin 1-beta (IL1-β), and tumor necrosis factor alpha (TNF-α). This leads to the activation of myeloid dendritic cells, through TLRs (toll-like receptor) receptors, which promote the production of IL12 and IL23, which regulate the differentiation and proliferation of helper T lymphocytes (Th1, Th17 and Th22). Th1 cells produce TNF-α and IFN-γ, Th2 cells produce IL22, while Th17 cells, in addition to producing TNF-α and IL22, also secrete IL17. These cytokines activate the hyperproliferation of keratinocytes in the epidermis and lead to the appearance of epidermal hyperplasia [7].

The treatment of moderate to severe psoriasis is essentially based on biological therapies (BTs) that inhibit the cytokines involved in the pathophysiology of the disease [8]. The first-line biologics are considered to be tumor necrosis factor inhibitors, also known as “anti-TNF” (infliximab [INF], etanercept [ETN], adalimumab [ADA] and certolizumab [CTL]), which specifically bind to TNF, inhibiting its interaction with TNF receptors, and therefore do not trigger the activation of dendritic cells. Moreover, the drug ustekinumab [UTK] acts by inhibiting the p40 subunit of IL-12 and IL-23, and consequently regulates the hyperproliferation of cytokine-producing T lymphocytes typical of psoriasis [3]. Second-line treatments include inhibitors of cytokines involved in the latter part of the typical psoriasis immune response, IL-17 (secukinumab, ixekizumab, brodalumab and the recently approved bimekizumab) and IL-23 (guselkumab, tildrakizumab and risankizumab) [9,10,11]. Morphological and functional alterations of receptors, associated proteins or these cytokines can modify the mechanism of action of BTs.

These drugs have proved to be highly effective and safe [12,13,14]. However, not all patients obtain good results, and loss of effectiveness is the main reason for interruption or change of BT, especially in the long term [15,16]. The statistical term “drug survival or persistence” describes the long-term success of the treatment, reflecting effectiveness and safety, as well as patient satisfaction with the therapy [17,18]. So far, studies have been published on survival of the various BTs from clinical practice records in patients diagnosed with psoriasis, and longer survival has been found with the IL-12 and IL-23 inhibitor drug (anti-IL-12/23, UTK) compared to anti-TNF drugs [19,20,21,22].

Despite the previous scientific evidence, the high percentage of switching of BTs in psoriatic patients represents a heavy economic burden for healthcare systems, as well as making medical decision-making difficult and worsening patient quality of life. Consequently, being able to determine the probability of response in advance has a positive impact on the healthcare system and on patients, making it possible to achieve a model of precision medicine [23]. Unhealthy lifestyle, development of inflammatory comorbidities, non-adherence to therapeutic regimens or genetic factors can contribute to a loss of efficacy of BTs. Although many of these factors have been studied, only genetic factors can be studied in a predictive manner before starting treatment. Therefore, pharmacogenetics may associate this variability in short- and long-term response with genetic factors, due to the fact that abnormalities in the genes involved in the pathological environment of the disease, metabolism or mechanism of action may influence the survival of these drugs [24].

Numerous pharmacogenetic studies in patients with psoriasis treated with BTs (specifically anti-TNF and UTK) have been conducted in the search for predictive biomarkers of effectiveness and/or toxicity. So far, it has been found that allelic variants and single-nucleotide polymorphisms (SNPs) in the human leukocyte antigen (HLA) genes, specifically *HLA-C* and *HLA-B/MICA*, which form part of the major histocompatibility complex and collaborate in identifying exogenous proteins that may trigger an immune response, could be associated with the response to BTs [25,26,27]. Similarly, SNPs in genes that code for proteins directly involved in the cytokine cascade, for example, the *TNF* gene in different positions (TNF-238, *TNF*-308, *TNF*-857, *TNF*-1031) or its receptor (*TNFRSF1B*), or even associated proteins (*TNFAIP3*, *TIRAP*). Additionally, SNPs in cytokine genes (*IL-1B*, *IL-6*, *IL-12B* and *IL-17*) or for receptors that regulate or trigger this cytokine cascade (*CD84*, *BCL2*,*CDKAL1*,*PGLYRP4-24*, *FCGR2A*, *FCGR3A*, *IL-17RA*, *IL-23R*, *SLCO1C1-PDE3A*,*TLR2*, *TLR5*, *TLR9* and *LY96*), which leads to hyperstimulation of the immune system and, consequently, excessive proliferation of keratinocytes in the epidermis, typical of psoriatic lesions, have proved to have an impact on the pharmacological response of BTs (Table 1) [28,29,30]. Although we know the possible impact of the above-mentioned biomarkers, it is currently not possible to transfer this information to everyday clinical practice because of the low level of evidence (small sample size studies, where the drugs studied are grouped and the results have not been confirmed with other studies). Therefore, the effect of these SNPs needs to be confirmed in further studies, so that they can be established as predictive biomarkers of response to the BTs indicated for psoriasis [31].

On the basis of all the foregoing, the object of this pharmacogenetic study was to evaluate the influence of polymorphisms in functional genes on the survival of anti-*TNF* and UTK drugs in patients diagnosed with moderate-to-severe psoriasis.

## 2. Results

### 2.1. Patient Characteristics

A total of 206 patients diagnosed with moderate-to-severe psoriasis who had been under treatment with anti-TNF or anti-IL12/23 (UTK) agents for at least 4 months were included in this study. Overall, data on 379 lines of treatment were collected, 247 with anti-TNF (ADA, ETN, INF, CTL) and 132 with UTK. The demographic, clinical and treatment variables are summarized in Table 2. The patients’ mean follow-up time was 7.8 years.

The mean age at the start of the study was 52.81 ± 14.17 years, with approximately equal proportions of men and women (women:104/198; 52.53%), and 35.42% (68/198) were of normal weight. Most of the patients had comorbidities (129/198; 65.15%) and 45% had psoriatic arthritis (89/198). The highest percentages of patients had vulgar or plaque psoriasis (87/198; 43.94%), with lesions mainly on the trunk and limbs (178/198; 89.9%) and a family history of psoriasis (115/198; 58.38%), and the mean age of diagnosis was 28 years (Appendix A).

The patients treated with UTK were younger (51.53 ± 13.13 years), had been diagnosed earlier (26 years), had a higher body mass index (BMI) (29.26 ± 6.13), and more of them had comorbidities than the other patients (94/132; 71.21%). In addition, the patients treated with UTK had a higher percentage of lesions in difficult-to-treat areas: nails (50%), flexures (37.12%) and genital area (16.28%). However, the bio-naive patients were generally treated with anti-TNF drugs (55.87%) as first-line therapy. Finally, there was more switching with anti-TNF drugs (78.14%), although the patients showed slightly greater adherence than those treated with UTK (80.58% vs. 74.81%) (Table 2).

### 2.2. Survival of Biological Therapies and Reasons for Discontinuation

The mean duration of BTs overall was 24 (9–51.5) months or 742 (288.5–1584) days (Table 2). The drug that achieved the longest survival was UTK (*p*_long rank_ = 0.07, 36 months vs. 24; Appendix A; Figure 1). Specifically, the patients treated with anti-TNF agents were 1.32 times more likely to discontinue the therapy than those treated with UTK (HR = 1.32; 95% CI = 1.02–1.7; *p* = 0.0316; Appendix A).

Next, we studied the reasons for discontinuation or switching of BTs (Appendix A). Loss of effectiveness was the main reason for discontinuation (188; 67.14%), followed by adverse events (55; 19.64%). The patients treated with UTK were 1.9 times more likely to interrupt treatment because of lack of effectiveness (OR = 1.895; 95% CI = 1.05–3.52; *p* = 0.024), whereas those treated with anti-TNF drugs interrupted the therapy because of remission of the disease (OR = 3.835; 95% CI = 1.05–21.07; *p* = 0.027) (Appendix A).

### 2.3. Influence of Clinical-Pathological Characteristics on Drug Survival

#### 2.3.1. Anti-TNF

Median pharmacological survival was higher in patients over the age of 54 (p_log-rank_ = 0.09; 24 vs. 17 months; Appendix A), bio-naive patients (p_log-rank_ = 0.007; 31 vs. 17 months for non-bio-naive patients) and those receiving concomitant treatment (p_log-rank_ = 0.02; 37 months for topicals vs. 29 for methotrexate vs. 19 for monotherapy vs. 9.5 for cyclosporine). In particular, older patients (95% CI = 0.97–0.99; *p* = 0.041) with a lower BMI (95% CI = 1.00–1.01; *p* = 0.023) and fewer systemic lines of treatment (95% CI = 1.01–1.19; *p* = 0.033) were less likely to interrupt treatment with anti-TNF drugs.

#### 2.3.2. Anti-IL12/23

Median survival of UTK was higher in patients who did not have psoriatic arthritis (p_log-rank_ = 0.003; 48 vs. 21 months; Appendix A) and in those being treated with concomitant therapy (p_log-rank_ = 0.03; 41 months for topicals vs. 37.5 for methotrexate vs. 23 for monotherapy vs. 11 for cyclosporine). In particular, patients with type II obesity (HR = 2.733; 95% CI = 1.30–5.73; *p* = 0.0078), those with psoriatic arthritis (HR = 1.899; 95% CI = 1.24–2.91; *p* = 0.0033) and those being treated with concomitant cyclosporine (HR = 4.457; 95% CI = 1.04–19.01; *p* = 0.043) were less likely to interrupt UTK therapy.

### 2.4. Genotype Distribution

Appendix A shows the minor allelic frequencies (MAFs) of all the SNPs; none were excluded from the statistical analysis (MAF > 1%). The genotype frequencies were in line with the expected values according to the Hardy–Weinberg equilibrium model, except in the *HLA*-C rs12191877 (*p* = 0.007), *TNF*-308 (rs1800629) (*p* = 0.04) and *IL17RA* rs4819554 (*p* = 0.01) gene polymorphisms (Appendix A). However, no statistically significant differences were found when they were compared with the frequencies described for the Iberian population (*HLA*-C rs12191877-T allele: 0.3175 vs. 0.1215, *p* = 0.73; *TNF*-308 [rs1800629]-A allele: 0.1263 vs. 0.1449, *p* = 0.96; *IL17RA* rs4819554-G allele: 0.2354 vs. 0.271 *p* = 0.95) [32]. The D’ and r2 linkage disequilibrium values are shown in Appendix A and Figure 2. The *IL1B* (rs1143623 and rs1143627), *TLR2* (rs4696480 and rs11938228) and *TNF* (rs361525 and rs1799964) were in strong linkage disequilibrium (Figure 2).

### 2.5. Influence of Gene Polymorphisms on Survival

#### 2.5.1. Anti-TNF

The bivariate analysis of survival of anti-TNF drugs showed an association between the *HLA*-C and *TNF* SNPs, specifically *TNF*-238, *TNF*-308 and *TNF*-1031 (Appendix A). The Kaplan–Meier anti-TNF survival curves for the *HLA*-C rs12191877 polymorphism are shown in Figure 3 (Figure 3a: CC vs. CT vs. TT genotypes; p_log-rank_ = 0.003; Figure 3b: T allele vs. CC; p_log-rank_ = 0.0007). Median survival for the TT and CT genotypes was 35 and 31 months, respectively (95% CI 16–not achieved [NA] and 23–40), whereas median survival for the CC genotype was only 14 months (11–24) (Appendix A).

Furthermore, an influence of the various polymorphisms of the TNF gene on increased survival of anti-TNF drugs was observed (Appendix A). Subsequently, the Cox multivariate regression confirmed the association between the *TNF*-1031 SNP and anti-TNF survival; however, in the *TNF*-238 and *TNF*-308 SNPs statistical significance was not maintained. The Kaplan–Meier anti-TNF survival curves for the TNF gene polymorphisms are shown in Appendix A (*TNF*-238 [rs361525], Appendix A: genotypic model; p_log-rank_ = 0.009, and Appendix A: A allele; p_log-rank_ = 0.007) and in Appendix A (*TNF*-308 [rs1800629], genotypic model; p_log-rank_ = 0.04).

Figure 4 shows the Kaplan–Meier survival curves for the *TNF*-1031 (rs1799964) SNP in patients treated with anti-TNF drugs (Figure 4a: genotypic model, p_log-rank_ = 0.003, and Figure 4b: C allele, p_log-rank_ = 0.004). Median survival for the *TNF*-1031 (rs1799964)-CT heterozygotic genotype was 44 months (25–70), greater than median survival for the TT and CC homozygotic genotypes, respectively (TT: 20 [15,16,17,18,19,20,21,22,23,24,25,26,27] months and CC: 20 [11–NA] months, Appendix A).

Finally, the Cox multivariate regression, adjusted for the patient’s age and bio-naivety, showed that the T allele of the *HLA*-C rs12191877 SNP (HR = 0.560; 95% CI = 0.40–0.78; *p* = 0.0006) and the C allele of the *TNF*-1031 (rs1799964) SNP (HR = 0.707; 95% CI = 0.50–0.99; *p* = 0.048) were associated with longer anti-TNF drug survival (Table 3).

#### 2.5.2. Anti-IL12/23

The bivariate survival analysis showed an association between the polymorphisms in the *IL1B*, *PDE3A*, *SLCO1C1*, *CD84* and *TLR5* genes, as well as in *HLA*-C, and longer UTK survival (Appendix A). The multivariate confirmed that the *PDE3A* rs11045392 and *SLCO1C1* rs3794271 polymorphisms were associated with longer UTK survival; however, statistical significance was not maintained in the *IL1B* rs1143627 polymorphism.

The Kaplan–Meier UTK survival curves for the *IL1B* rs1143627 polymorphism are in Appendix A (Appendix A: genotypes, p_log-rank_ = 0.03, and Appendix A: A vs. GG allele, p_log-rank_ = 0.03; Appendix A), and the median survival corresponding to patients carrying the GG genotype was 67 (67–NA) months, whereas for the AG and AA genotypes it was 27 (18–40) months and 40 (23–78) months, respectively. On the other hand, Figure 5 shows the Kaplan–Meier curve for the *PDE3A* rs11045392 and *SLCO1C1* rs3794271 polymorphisms (Figure 5a: CC vs. CT vs. TT genotype, p_log-rank_ = 0.04, and Figure 5b: T vs. CC allele, p_log-rank_ = 0.02), and the median survival for the TT genotype was 48 (23–NA) months. It was 37 (26–94) months for heterozygotic and 25 (18–40) months for CC homozygotic patients (Appendix A).

Moreover, the Kaplan–Meier curves for the *HLA*-C and *TLR5* polymorphisms are in the Appendix A (Appendix A: *HLA-C* rs12191877-T allele, p_log-rank_ = 0.06; Appendix A: genotype, p_log-rank_ = 0.07). Median survival for the *HLA*-C rs12191877-T allele was 36 (21–112) months, while in *CD84* rs6427528-GG it was 40 (27–66) months, 23 (18–40) for AG and 10 (NA–NA) in the case of patients carrying the AA genotype, respectively. Finally, median UTK survival in *TLR5* rs5744174-G was 41 (26–67) months, compared to 34 months for patients carrying the AA genotype (Appendix A).

Finally, the Cox multivariate regression analysis confirmed the association between UTK survival and the *TLR5* rs5744174-G (HR = 0.589; 95% CI = 0.37–0.92; *p* = 0.02), *CD84* rs6427528-A (HR = 0.557; 95% CI = 0.35–0.88; *p* = 0.013) and PDE3A rs11045392 together with *SLCO1C1* rs3794271-T (HR = 0.508; 95% CI = 0.32–0.79; *p* = 0.002) SNPs, adjusted for the psoriatic arthritis variable (Table 4).

## 3. Discussion

The biological therapies indicated for the treatment of moderate-to-severe psoriasis are highly effective and safe; however, certain patients experience a short-term lack of response and/or a gradual long-term loss of response [15]. This variability in response may be due to genetic factors, because abnormalities in the genes involved in the pathological environment of the disease, metabolism or mechanism of action may influence the effectiveness of BTs. Specifically, polymorphisms in the genes that encode certain human leukocyte antigens, cytokines, receptors and transporters related to the immune system have proved to play a crucial role in interindividual variability in the response to anti-TNF drugs and UTK [25,26,28,31]. In this study, 379 lines of treatment (anti-TNF: 247 and UTK: 132) in 198 patients diagnosed with moderate-to-severe psoriasis were evaluated to determine the influence of SNPs in functional genes on drug survival.

Firstly, we evaluated the survival of the drugs studied and observed that UTK showed longer survival than the anti-TNF drugs (3 vs. 2 years; p_Log rank_ = 0.07; Figure 1; Appendix A), the main reason for discontinuation being loss of effectiveness of UTK (OR = 1.895; 95% CI = 1.05–3.52; *p* = 0.024; Appendix A). Similarly, the various systematic reviews and meta-analyses carried out to date have observed longer survival of UTK compared to anti-TNF [20,22].

On the other hand, we evaluated the impact of the SNPs in functional genes selected for this study on the survival of the BTs used in psoriasis, finding that patients carrying the T allele of the *HLA*-C rs12191877 polymorphism and those with the C allele of the *TNF*-1031 (rs1799964) polymorphism, adjusted for age and bio-naivety, were associated with longer pharmacological survival of anti-TNF drugs, and that patients who had not developed psoriatic arthritis, and those carrying the G allele of the *TLR5* rs5744174 polymorphism, the A allele of the *CD84* rs6427528 polymorphism or the T allele of the *PDE3A* rs11045392 polymorphism, which was in linkage disequilibrium with *SLCO1C1* rs3794271, were associated with longer survival of UTK.

The association observed in the patients in this study carrying the T allele of the *HLA*-C rs12191877 SNP with longer survival of anti-TNF drugs, compared to patients carrying the wild-type homozygous (CC) genotype (T vs. CC; HR = 0.56; 95% CI = 0.40–0.78; *p* = 0.0006) is in line with previous studies [25]. Human leukocyte antigens (HLAs) form part of the major histocompatibility complex (MHC) and collaborate in identifying exogenous proteins that can trigger an immune response [33,34]. Specifically, the HLA-C glycoprotein promotes the production of cytokines and activates monocytes. High HLA-C levels have been related to the risk of suffering from inflammatory bowel disease or spondyloarthropathy [35]. Consequently, multiple pharmacogenetic studies have been conducted to investigate the impact of HLA genetic variants on the development of psoriasis and its response to the various BTs [25,27,36]. In particular, the *HLA*-C rs12191877 polymorphism was associated with the response to anti-TNF drugs (a reduction of 75% in psoriasis area and severity index score [PASI 75] at 3 months) in 144 white patients (from Spain) (OR = 0.30, 95% CI = 0.11–0.88, *p* = 0.027) [37].

Moreover, this SNP is in linkage disequilibrium with the *HLA-C*0602* allele (also known as the *HLA-Cw*06* allele) [38]. The presence of this allele has proved to confer a high risk of suffering from psoriasis, but its association with the response to anti-TNF drugs has yet to be confirmed [25,26,28,39,40]. In two studies with white patients it was observed that patients with the *HLA-Cw*06* allele showed a higher risk of not responding to anti-TNF therapy (109 Spanish patients treated with ADA, ETN or INF, PASI75 in week 24, OR = 58.1, 95% CI = 71.7–93.8, *p* = 0.049; 1326 British patients treated with ADA, PASI90 at 6 months, OR = 2.95, *p* < 0.0001) [27,41]. However, there is disagreement with other studies published by Ryan et al., Talamonti et al. and Caldarola et al., because no statistically significant association was found for this allele (*p* > 0.05) [39,40,42].

Our study observed an association between the C allele of the *TNF*-1031 (rs1799964) polymorphism of the tumor necrosis factor gene promoter region and survival of anti-TNF (HR = 0.707; 95% CI = 0.50–0.99; *p* = 0.048, C vs. TT) [28]. The importance of this SNP lies in the proinflammatory cytokine encoded by the gene in question (TNF), which is directly associated with the pathogenesis of psoriatic lesions, as well as with the target of four biological drugs (ADA, ETN, INF and CTL). TNF promotes the production of T cells, which trigger local proliferation of epidermal keratinocytes, and therefore genetic abnormalities of this cytokine can directly influence the functioning of anti-TNF drugs [43,44,45]. However, there is a previous study that evaluated the impact of this polymorphism and that associates its TT genotype with a greater short-term response to anti-TNF (109 Spanish white patients who reached PASI75 in month 3, TT: 90.8% vs. CC + CT genotype: 75.7%, *p* = 0.047; 85.5% vs. 65.7% in month 6, *p* = 0.038). Specifically, INF attained the greatest response at 3 months (PASI75: 84.2% vs. 42.9%, *p* = 0.024; PASI90: 73.7% vs. 28.6, *p* = 0.015) and at 6 months (PASI75: 94.1% vs. 53.8%; *p* = 0.025; PASI90: 76.5% vs. 30.8%, *p* = 0.025; ΔPASI: 94.1% vs. 64.7%, *p* = 0.019) [41].

With regard to treatment with UTK, it was found that patients carrying the G variant allele of the *TLR5* rs5744174 SNP showed longer survival of UTK than those carrying the wild-type AA genotype (G allele vs. AA: HR = 0.589, 95% CI = 0.37–0.92, *p* = 0.02). These results are in line with the study conducted by Loft and his team on 230 white patients (from Denmark) with psoriasis treated with UTK. It was observed that the patients carrying the *TLR5* rs5744174-C allele (complementary to the G allele) showed a better response (ΔPASI after 3 months) to UTK treatment (OR = 5.26, 95% CI = 1.93–14.38, *p* = 0.0012, q = 0.19) [30]. The *TLR5* gene, belonging to the toll-like receptor (TLR) family, codes for a transmembrane protein in which the toll interleukin receptor (TIR)-type intracellular domain binds to IL-1, generating an inflammatory cascade when the exterior domain recognizes bacterial flagellin [46,47]. Therefore, when the variant allele is present in the *TLR5* rs5744174 polymorphism, IL-1 levels decrease and CCL20 and IFN-γ levels increase, reducing the inflammatory response [48,49,50].

The effect of the *CD84* rs6427528 receptor gene polymorphism on the response of autoimmune diseases to BTs has been studied. In particular, a meta-analysis of genome-wide association studies (GWAS) with white patients (13 studies/2706 patients) diagnosed with rheumatoid arthritis and treated with anti-TNF showed that the *CD84* rs6427528-AG genotype was associated with greater effectiveness of treatment with ETN (733) (*p* = 0.004) [51]. In psoriasis, a pharmacogenetic study has been conducted in a white population (from the Netherlands) in which the *CD84* rs6427528-AG genotype was associated with a better response (ΔPASI at 12 weeks) to treatment with ETN (25 patients with 161 ETN episodes; beta = −2.028, 95% CI = −3.794–0.261, *p* = 0.025) [29]. Our study was unable to confirm this association, but it was observed that the GG genotype of the *CD84* rs6427528 polymorphism was associated with UTK survival (GG vs. A: HR = 0.557; 95% CI = 0.35–0.88; *p* = 0.013). These results are interesting, since genetic abnormalities in the *CD84* gene, belonging to the CD2 superfamily of cell surface receptors, even in the non-coding region (3′UTR), mean greater expression of this gene in mononuclear peripheral blood cells, and consequently deregulation of T and B cell signaling, as well as in the adhesion and activation of immune cells [29,52].

Finally, the *PDE3A* rs11045392 SNP, which is in linkage disequilibrium with *SLCO1C1* rs3794271, showed a statistically significant association between the T allele or the TT and CT genotypes, compared to the CC genotype, and UTK survival (T allele vs. CC: HR = 0.508; 95% CI = 0.32–0.79; *p* = 0.002). In psoriasis, a study evaluated the influence of the *SLCO1C1* rs3794271 and *PDE3A* rs11045392 polymorphisms on anti-TNF response in 130 white patients (from Spain) and showed that patients carrying the C allele obtained a better response (ΔPASI after 3 months) (*p* = 0.00057) [53]. This may be due to the fact that both genes are related to the physiopathogenesis of psoriasis and therefore to the response to various BTs. Specifically, the *PDE3A* gene is expressed mainly in cardiac tissue and encodes a phosphodiesterase, which is responsible for internal control of nucleotide signaling, while the *SLCO1C1* gene encodes a sodium-independent transporter with high affinity for the thyroid hormone in brain tissue.

Among the main limitations of our study that should be highlighted is the limited sample size, which may be responsible for the lack of statistical association found between the SNPs and pharmacological survival. Nevertheless, we obtained important and representative results which show the influence that certain genetic polymorphisms have on the probability of changing treatment with anti-TNF drugs or UTK in patients with moderate-to-severe psoriasis. The fact that the four anti-TNF drugs (ADA, ETN, INF and CTL) were grouped together in the statistical analysis may seem to be a limitation; however, grouping them in this way increases the statistical power, and they are usually studied together. In addition, it should be pointed out that this study was carried out with patients from only two hospitals, all diagnosed with moderate-to-severe psoriasis, following the same therapeutic protocols, by the same team of dermatologists. This makes it possible to achieve a high degree of homogeneity in the cohort, as well as in the process of collecting the variables.

To sum up, the results suggest that the *HLA*-C rs12191877 and *TNF*-1031 (rs1799964) SNPs are associated with anti-TNF drug survival, while *TLR5* rs5744174, *CD84* rs6427528 and *PDE3A* rs11045392 together with *SLCO1C1* rs3794271 are associated with UTK survival. Consequently, these polymorphisms could act as survival biomarkers for the various BTs in patients with moderate-to-severe psoriasis and could be useful for direct selection of the appropriate biological therapy, thereby achieving more individualized medicine with lower pharmaceutical cost and better results for the patient. Although more studies in larger cohorts will be needed to confirm the prognostic value of the biomarkers, the information on the persistence, safety and tolerability of treatment in the short and long term may help patients and healthcare professionals in making decisions to begin treatment with a BT [18].

## 4. Materials and Methods

### 4.1. Study Design

We conducted a prospective and retrospective observational cohort study.

### 4.2. Ethics Statements

This study was carried out with the approval of the Ethics and Research Committee of the Sistema Andaluz de Salud (SAS: Andalusian Health Service) and in accordance with the Declaration of Helsinki (code: FG-DERM-SNP-0702-N-21, 27 October 2015). The patients signed a written informed consent form for collection of saliva samples and their donation to the biobank. The samples were coded and treated confidentially.

### 4.3. Study Population

The study included 260 white patients diagnosed with modern-to-severe psoriasis treated with anti-TNF and UTK drugs, recruited at the Hospital Universitario Virgen de las Nieves, Granada, Spain, and at the Dermatology Clinic, Department of Clinical, Internal, Anesthesiological and Cardiovascular Sciences, La Sapienza Università di Roma, Italy, during the period from January 2019 to November 2021 (Figure 6). The inclusion criteria for the group of patients were age 18 years or more, diagnosis of moderate-to-severe psoriasis (body surface area [BSA] and/or PASI > 10) in treatment with BT (ADA, INF, ETN, CTL or UTK) and clinical data available. The patients were treated in accordance with the EuroGuiDerm Guideline [54].

### 4.4. Socio-Demographic and Clinical Variables

The socio-demographic information, including sex, age and BMI, was collected from the patients’ medical records at the beginning of the study. We collected the clinical features of the psoriasis, including age at diagnosis of psoriasis, family history of psoriasis, baseline PASI, type of psoriasis (vulgaris or plaque, pustular, inverse, guttate or a combination of 2, 3 or 4 types) and location of the lesions, according to the consensus document on evaluation and treatment of moderate-to-severe psoriasis: Psoriasis Group of the Spanish Academy of Dermatology and Venereology [3]. We also studied the patients’ comorbidities: psoriatic arthritis, hypertension, dyslipidaemia, and others, such as diabetes mellitus, other autoimmune diseases, history of cancer, respiratory diseases (asthma, chronic obstructive pulmonary disease [COPD]), mental health disorders and metabolic syndrome. In addition, we studied variables related to the lines of treatment, such as bio-naivety, the line of treatment in general (systemic and biological treatments), the reason for the interruption of the biological therapy, concomitant treatment, therapeutic adherence and duration of biological therapy in months and in days. Therapeutic adherence was evaluated with objective measures (dispensing percentage in the biological therapy record) and subjective measures (Morisky–Green and Haynes–Sackett tests).

### 4.5. Sample Processing and Genotyping

#### 4.5.1. DNA Isolation

After the patients’ inclusion and signing of the informed consent, saliva samples were collected with buccal swabs (OCR-100 Kit from BD, Plymouth, UK). The DNA was extracted using the QIAamp DNA Mini extraction kit (Qiagen GmbH, Hilden, Germany), following the specifications provided by the manufacture for purifying DNA from saliva, and stored at −20 °C. The DNA concentration and purity were measured using a NanoDrop 2000 UV spectrophotometer with the absorbance ratio at 280/260 and 280/230.

#### 4.5.2. Detection of Gene Polymorphisms and Quality Control

Table 5 shows the 29 SNPs that were genotyped by real-time polymerase chain reaction (PCR) allelic discrimination assay using TaqMan probes (ABI Applied Biosystems, QuantStudio 3 Real-Time PCR System, 96 wells), following the manufacturer’s instructions. In the analysis of HLA-B/MICA and BCL2, customized probes by ThermoFisher Scientific (Waltham, MA, USA) were used.

To assess the internal consistency of the genotyping, 10% of the results were confirmed by Sanger sequencing analysis. The real-time PCR and the Sanger sequencing were performed at the Pharmacogenetics Unit of the Hospital Universitario Virgen de las Nieves, Granada, Spain. Subsequently, quality control to eliminate rare variants and samples with missing data of the genotypes was carried out. The criteria for SNP quality control were (1) missing genotype rate per SNP < 0.05; (2) minor allele frequency > 0.01; (3) *p*-value > 0.05 in the Hardy–Weinberg equilibrium test; (4) missing genotype rate between cases and control < 0.05.

### 4.6. Survival Variables

The data on treatment with anti-TNF and UTK biological therapies were recorded, including the dates of starting and finishing the therapy and the reasons for discontinuation. When treatment with a drug is discontinued, a new line of treatment with another drug is initiated. Therefore, the survival of the drug is defined as the time from initiation to discontinuation of treatment with the particular BT. The data were obtained from the patients’ medical records.

### 4.7. Statistical Analysis

First, the study sample size was calculated to obtain a precision of 1 unit in the estimate of the logarithm of an odds ratio using a two-sided 95% normal asymptotic confidence interval, based on previous studies.

The patient characteristics were collected and presented as frequencies with percentages for the qualitative variables. The quantitative variables were expressed as mean (plus or minus standard deviation) for the variables with normal distribution or medians and percentiles (25 and 75) for those with non-normal distribution. The Lilliefors (Kolmogorov–Smirnov) test was used to evaluate normality.

We used the Kaplan–Meier survival curve with the log-rank test to compare survival for patients treated with different BTs and the associations of demographic, clinical and genetic variables. The association with SNPs was evaluated in multiple models (genotypic, dominant and recessive), which were defined as follows: genotypic (DD vs. Dd vs. dd), dominant ((DD, Dd) vs. dd) and recessive (DD vs. (Dd, dd)), where D is the major (wild-type) allele and d the minor (variant) allele. Moreover, a univariate and multivariate Cox regression analysis was performed by calculating the hazard ratio (HR) and the 95% confidence interval (95% CI) to predict the risk factors of discontinuing treatment with BTs. For the multivariate analysis, we used the Cox proportional risks regression model, taking into account the variables associated with statistically significant results in the bivariate analysis of drug survival (backward stepwise selection method).

All the tests were two-sided, with a significance level of *p* < 0.05. The data analysis was performed with the R 4.0.2 program [55]. The Hardy–Weinberg equilibrium and the linkage disequilibrium were determined with the D’ and r2 coefficients and estimated using the PLINK, Haploview 4.2 and SNPStat programs [56,57,58].

## 5. Conclusions

Our results suggest that patients with moderate-to-severe psoriasis under treatment with TNF inhibitor drugs or UTK show greater pharmacological survival of UTK. Furthermore, various genetic polymorphisms associated with BT survival have been found. Patients carrying the T allele of the *HLA-C* rs12191877 SNP or the C allele of the *TNF*-1031 (rs1799964) SNP are associated with greater pharmacological survival of anti-TNF drugs, while patients who have not developed psoriatic arthritis or are carriers of the G allele of the *TLR5* rs5744174 polymorphism, the A allele of *CD84* rs6427528 or the T allele of *PDE3A* rs11045392, which is in linkage disequilibrium with *SLCO1C1* rs3794271, are associated with longer UTK survival. Consequently, these results confirm that these genetic polymorphisms could be useful as pharmacological survival biomarkers for biological therapies indicated for psoriasis, increasing the current scientific evidence. This information will make it possible to implement personalized medicine that would reduce economic healthcare costs, facilitate medical decision-making and improve patient quality of life. However, further pharmacogenetic studies need to be conducted to confirm these associations.

## Figures and Tables

**Figure 1 ijms-24-08703-f001:**
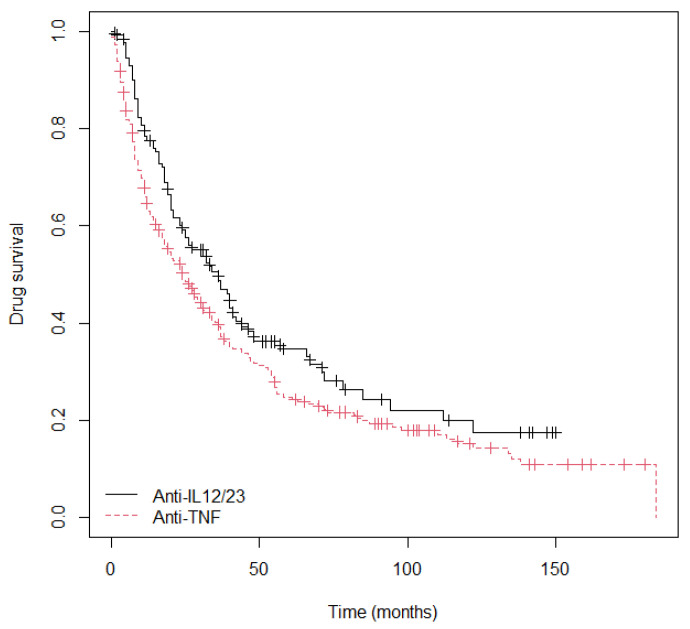
Drug survival all treatments by target.

**Figure 2 ijms-24-08703-f002:**
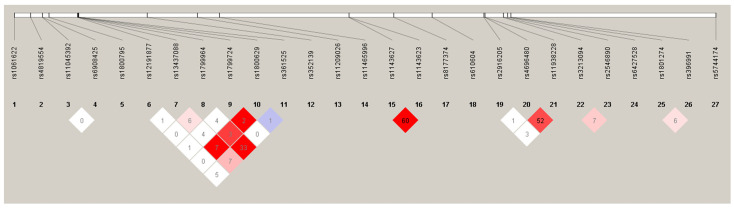
Linkage disequilibrium. The different shades of red represent the values of D, red being the highest value and white the lowest.

**Figure 3 ijms-24-08703-f003:**
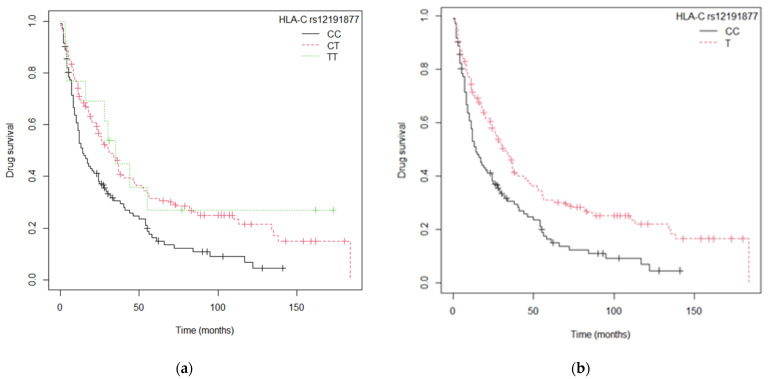
The Kaplan–Meier anti-TNF survival curves for the *HLA-C* rs12191877 polymorphism: (**a**) *HLA-C* rs12191877 genotypic model (CC vs. CT vs. TT); (**b**) *HLA-C* rs12191877 model (T allele vs. CC).

**Figure 4 ijms-24-08703-f004:**
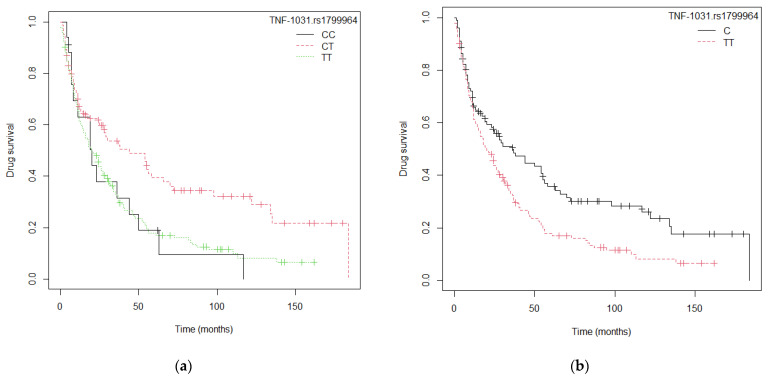
The Kaplan–Meier survival curves for the *TNF*-1031 rs1799964 SNP in patients treated with anti-TNF drugs: (**a**) *TNF*-1031 rs1799964 genotypic model (CC vs. CT vs. TT); (**b**) *TNF*-1031 rs1799964 model (Allele C vs. TT).

**Figure 5 ijms-24-08703-f005:**
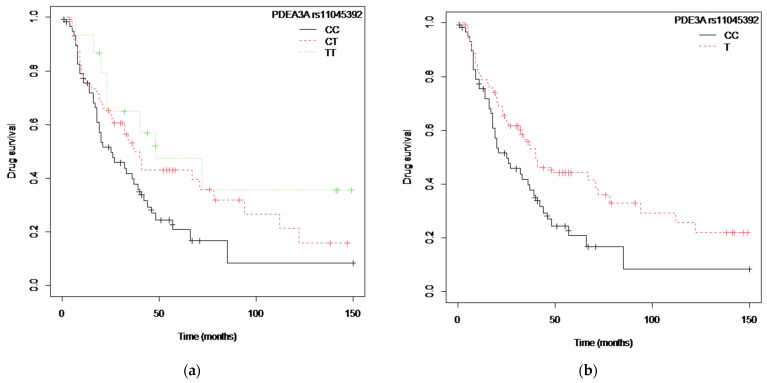
The Kaplan–Meier curve for the *PDE3A* rs11045392 and *SLCO1C1* rs3794271 polymorphisms: (**a**) *PDE3A* rs11045392 and *SLCO1C1* rs3794271 (CC vs. CT vs. TT genotype); (**b**) *PDE3A* rs11045392 and *SLCO1C1* rs3794271 model (Allele T vs. CC).

**Figure 6 ijms-24-08703-f006:**
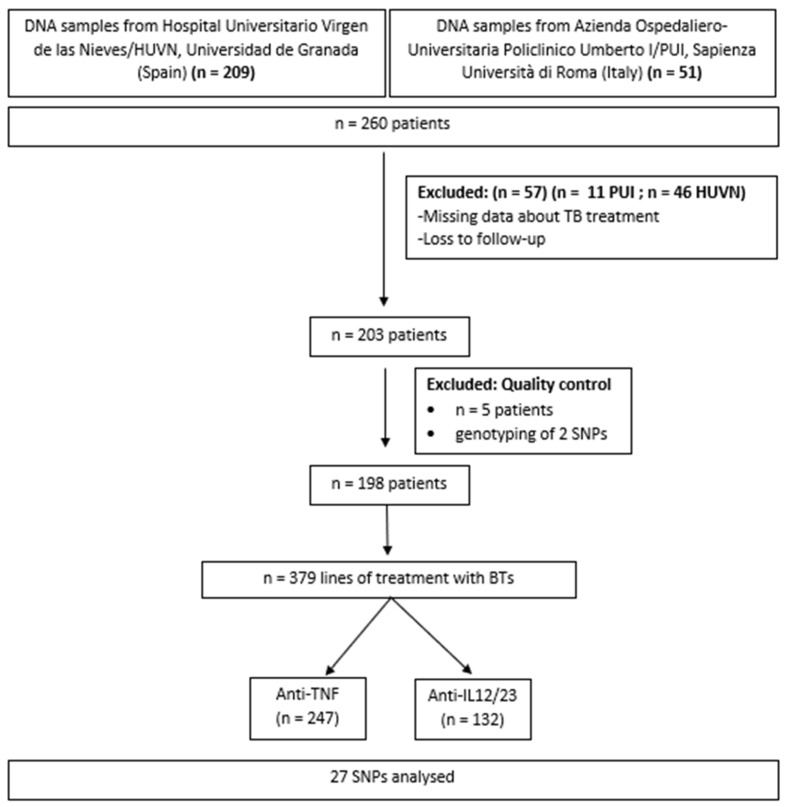
Selection of patients included in the study.

**Table 1 ijms-24-08703-t001:** List of abbreviations for other terms used in the introduction.

ADA	Adalimumab
ANTI-IL12/23	Il-12 and Il-23 inhibitor drug
ANTI-TNF	Tumor necrosis factor inhibitors drugs
BCL2	Apoptosis regulator Bcl-2
BTs	Biological therapies
CD84	CD84 receptor
CDKAL1	CDK5 regulatory subunit associated protein 1 like 1
CTL	Certolizumab
ETN	Etanercept
FCGR	Fc gamma receptor
HLA	Human leukocyte antigen
IL	Interleukins
IL-17RA	Interleukins 17-A receptor
IL-23R	Interleukins 23 receptor
INF	Infliximab
LY96	Lymphocyte antigen 96
PDE3A	Fosfodiesterase 3A
*PGLYR*	Peptidoglycan recognition protein 4
SLC	Solute carrier family
SNPs	Single-nucleotide polymorphisms
TIRAP	TIR domain containing the Adaptor Protein
TLR	Toll-like receptor
TNF	Tumor necrosis factor
TNFAIP3	Tumor necrosis factor alpha-induced protein 3
UTK	Ustekinumab

**Table 2 ijms-24-08703-t002:** Description of Treatments.

Variable	All Treatments (379)		Anti-TNF (247)		Anti-IL12/23 (132)	
	N	% or Mean ± SD	N	% or Mean ± SD	N	% or Mean ± SD
**Female**	199	51.61	122	49.39	77	58.33
**Age at baseline**	324	54 (42.75–63)	197	54.29 ± 13.45	127	51.53 ± 13.13
**BMI at baseline**	376	27.68 (24.22–31.47)	245	27.42 (24.22–30.48)	131	29.26 ± 6.13
Normal weight	114	30.89	78	32.37	36	28.12
Overweight	132	35.77	90	37.34	42	32.81
Obesity Type I	76	20.60	45	18.67	31	24.22
Obesity Type II	29	7.86	17	7.05	12	9.38
Obesity Type III	18	4.88	11	4.56	7	5.47
**Comorbidities**						
Psoriatic arthritis	192	50.66	129	52.23	63	47.73
Hypertension	137	36.15	96	38.87	41	31.06
Dyslipidaemia	168	44.33	111	44.94	57	43.18
Other comorbidities	254	67.02	160	64.78	94	71.21
A**ge at PS diagnosis**	379	28 (18–42)	247	29 (18.5–42.5)	132	26 (16–39.25)
**Family history of PS**	210	55.85	133	54.51	77	58.33
**Type of PS**						
Plaque	352	92.88	231	93.52	121	91.67
Other types of PS	27	7.12	16	6.48	11	8.33
**Location of lesions**						
Trunk and limbs	347	91.56	227	91.9	120	90.91
Scalp and face	251	66.23	153	61.94	98	74.24
Nails	162	42.74	96	38.87	66	50
Palmoplantar	62	16.36	40	16.19	22	16.67
Flexures	117	30.87	68	27.53	49	37.12
Genital	45	12.03	24	9.8	21	16.28
**Bio-naive**	182	48.02	138	55.87	44	33.33
**Lines of treatment**	379	3 (2–4)	247	3 (2–4)	132	4 (3–5)
**Discontinuation of BT**						
No	99	26.12	54	21.86	45	34.09
Yes	280	73.88	193	78.14	87	65.91
**Discontinuation of BT and reason**	280	73.88				
Ineffective in the short term	36	9.50	26	10.53	10	7.58
Ineffective in the long term	152	40.11	95	38.46	57	43.18
Remission therapy	17	4.49	15	6.07	2	1.52
Adverse events	56	14.78	42	17	13	9.85
Other reasons	19	5.01	15	6.07	5	3.79
**Concomitant treatment for PS**						
Acitretin	2	0.53	2	0.81	–	–
Cyclosporine	6	1.58	4	1.62	2	1.52
Methotrexate	52	13.72	32	12.96	20	15.15
Topicals	102	26.91	55	22.27	47	35.61
**Adherent to BT**						
Yes	293	78.55	195	80.58	98	74.81
No	80	21.45	47	19.42	33	25.19
**Baseline PASI**	239	8.2 (5–10.7)	132	9.1 (5.17–12)	107	7.2 (4–10)
**Duration of BT**						
Months	379	24 (9–51.5)	247	21 (8–52.5)	132	28.5 (13.25–50.25)
Days	379	742 (288.5–1584)	247	666 (249–1603)	132	884.5 (397.5–1506.75)

Anti-TNF: tumor necrosis factor inhibitor (Adalimumab, Certolizumab pegol, Etanercept and Infliximab); anti-IL12/23: interleukin 12 and interleukin 23 inhibitor (Ustekinumab); BMI: body mass index; BT: biological therapy; ineffective in the short term (BT < 6 months) and ineffective in the long term (BT > 6 months); PASI: psoriasis area severity index score; PS: psoriasis. Qualitative variables are shown as numbers (percentage, %). Shapiro-Wilk normality test or Lilliefors (Kolmogorov-Smirnov) normality test. Quantitative variables with a normal distribution are shown as mean ± standard deviation. Quantitative variables with a non-normal distribution are shown as *p*_50_ (*p*_25–_*p*_75_).

**Table 3 ijms-24-08703-t003:** Influence of clinical characteristics and gene polymorphisms on drug survival in 247 psoriasis patients treated with anti-TNF agents.

	Anti-TNF Drug Survival		
	HR	95% CI	*p-*Value
Age at baseline	0.981	0.96–0.99	0.004
Bio-naïve	0.653	0.46–0.91	0.013
*HLA-C* rs12191877 (T vs. CC)	0.560	0.40–0.78	0.0006
*TNF*-1031 (rs1799964) (C vs. TT)	0.707	0.50–0.99	0.048

HR = hazard ratio; CI: confidence interval.

**Table 4 ijms-24-08703-t004:** Influence of clinical characteristics and gene polymorphisms on drug survival in 132 psoriasis patients treated with anti-IL12/23 (UTK).

	Anti-IL12/23 Drug Survival		
	HR	95% CI	*p-*Value
Psoriatic arthritis	2.526	1.61–3.96	0.00005
*TLR5* rs5744174 [G vs. AA]	0.589	0.37–0.92	0.02
*CD84* rs6427528 [GG vs. A]	0.557	0.35–0.88	0.013
*PDE3A* rs11045392; *SLCO1C1* rs3794271 [T vs. CC]	0.508	0.32–0.79	0.002

HR: hazard ratio; CI: confidence interval.

**Table 5 ijms-24-08703-t005:** Information on the TaqMan SNP genotyping assay used in our study.

Gene	dbSNP ID	Taqman ID	Gene	dbSNP ID	Taqman ID
*BCL2*	rs59532114	ANFVZRF *	*PDE3A*	rs11045392	C__31106576_10
*CD84*	rs6427528	C__29332612_10	*PGLYRP4*-24	rs2916205	C___9092093_10
*CDKAL1*	rs6908425	C___2504037_10	*SLCO1C1*	rs3794271	C__27502188_10
*FCGR2A*	rs1801274	C___9077561_20	*TIRAP*	rs8177374	C_25983622_10
*FCGR3A*	rs396991	C__25815666_10	*TLR2*	rs4696480	C__27994607_10
*HLA-B/MICA*	rs13437088	ANPRYYF *	*TLR2*	rs11938228	C__32212770_10
*HLA-C*	rs12191877	C_176062476_10	*TLR5*	rs5744174	C_25608809_10
*IL12B*	rs3213094	C__29927086_10	*TLR9*	rs352139	C___2301953_10
*IL12B*	rs2546890	C__15894458_10	*TNFAIP3*	rs610604	C____884105_20
*IL17RA*	rs4819554	C____337392_30	*TNFRSF1B*	rs1061622	C___8861232_20
*IL1B*-623	rs1143623	C__1839941_10	*TNF*-1031	rs1799964	C___7514871_10
*IL1B*-627	rs1143627	C__1839944_10	*TNF*-238	rs361525	C___2215707_10
*IL23R*	rs11209026	C___1272298_10	*TNF*-308	rs1800629	C__7514879_10
*IL6*	rs1800795	C__1839697_20	*TNF*-857	rs1799724	C__11918223_10
*LY96*	rs11465996	C__30755344_10			

dbSNP ID: identifier of Single Nucleotide Polymorphism Database. SNPs that were not included in the statistical analysis, because they did not pass quality control, are shaded in grey. * These SNPs were analyzed using custom assays by ThermoFisher Scientific (Waltham, MA, USA).

## Data Availability

Data unavailable due to privacy and ethical restrictions.

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
