# Peer review of "Impact of Functional Polymorphisms on Drug Survival of Biological Therapies in Patients with Moderate-to-Severe Psoriasis"

_ijms, 2023, doi:10.3390/ijms24108703_

Round 1

Reviewer 1 Report

General Comment

The section should be rearranged in the following standard research format;

1.       Introduction

2.       Material and Methods

3.       Result

4.       Discussion

5.       Conclusion

I recommend that the authors clarify how their findings add to the current body of literature and what specific clinical applications these findings may have. Providing additional context and clarity could help strengthen the conclusion section and provide greater value to readers

Abstract:

The manuscript abstract concisely summarizes the study’s background, objectives, methods, results, and conclusions. However, it would be helpful to include information on the study’s limitations and any implications for future research.

Introduction:

The introduction provides a strong foundation for the study and adequately justifies the need for the research. The authors have provided a comprehensive overview of the background of the study, the importance of the research question, and the current state of knowledge in the field. The specific comments provided are intended to improve the clarity and context of the introduction and enhance its impact.

1.       Provide a brief overview of the pathophysiology of psoriasis, especially as it relates to the use of BTs. This would provide a better understanding of the role of BTs in treating psoriasis and the need for identifying predictive biomarkers of response.

2.       Discuss the mechanism of action of these drugs, especially as it relates to the genes and proteins involved in the cytokine cascade. This would provide a better understanding of the potential impact of genetic variations on drug response.

3.       Provide a brief overview of the factors contributing to the loss of effectiveness and the need to change or discontinue BTs. This would help to contextualize the importance of identifying predictive biomarkers of response.

4.       Briefly discuss the rationale for using pharmacogenetics in identifying biomarkers of response and how it differs from other approaches, such as clinical or demographic factors.

5.       Provide a brief overview of the limitations of numerous pharmacogenetic studies investigating the association between genetic variations and response to BTs in psoriasis, especially in terms of their sample sizes, study designs, and statistical power. This would provide a better understanding of the need for further research in this area and the limitations of current evidence.

6.       You have provided a clear statement of the research question and the aim of the study. However, it would be helpful if the authors could briefly overview the specific polymorphisms that will be investigated in the study and their potential impact on drug survival. This would provide a better understanding of the specific research question and the study’s potential impact on clinical practice.

7.       Could provide a list of abbreviations for other terms used in the introduction, such as SNPs and HLA.

Material and Methods

1.       Materials and Methods section is well-written and provides sufficient detail for readers to understand the study design, recruitment, data collection, and sample processing. However, offer more information on the quality control procedures for genotyping.

2.       The study’s sample size is relatively small, and the findings must be confirmed in larger populations.

Result and Discussion

1.       It will be ok to incorporate investigating the impact of other potential factors that may influence the effectiveness of biological therapies, such as lifestyle factors and comorbidities.

Reviewer 2 Report

The paper discusses the role of genetic polymorphisms in the psoriasis patient population in response to biologic drug treatment.

Specifically, both certain clinical characteristics, such as age, type of psoriasis, bio-naive status, and minor allelic frequencies of SNPs are researched. Finally, correlations between the above characteristics and drug survival of therapy with TNF-inhibitors and usekinumab are analyzed. 

The topic is interesting, current and in line with published studies. It adds useful elements for further works on precision medicine.

The work is well illustrated, and the conclusions are consistent and coherent with the results. The materials and methods conducted are well explained, and the statistics are clear.

English is fluent.

Reviewer 3 Report

This study is really important, and I recommend to publish it after following questions clarification by authors.

1. Page 7, Figure 2: Legend is on up of the figure, please correct it.

2. Page 1, line 16: Please include the full name of ALA. Don’t use abbreviation only in abstract.

3. In pages 7 and 8, there are Figures 3 and 6. Where are Figures 4 and 5.

4. Also, in between Figures 6 and 9, Figures 7 and 8 are missing.

5. Page 9, Figure 9: ‘(a)’ covers note of x-axis, please correct it.

Round 2

Reviewer 1 Report

The following comments I made were not implemented.

Re-arrange the sections into Introduction, Material and Method or Methodology, Result, Discussion, and Conclusion. However, the sections remain the same. Introduction, Result, Discussion, and Conclusion. 

Also, the mechanism of action of these drugs, especially as it relates to the genes and proteins involved in the cytokine cascade, was not included as recommended.

Reviewer 3 Report

To me, this manuscript is acceptable.

Author Response

We appreciate your comments, they have been useful to improve the manuscript.